# In Vitro Time-Kill of Common Ocular Pathogens with Besifloxacin Alone and in Combination with Benzalkonium Chloride

**DOI:** 10.3390/ph14060517

**Published:** 2021-05-27

**Authors:** Joseph Blondeau, Heleen DeCory

**Affiliations:** 1Clinical Microbiology, Royal University Hospital, Saskatoon, SK S7N 0W8, Canada; 2Bausch + Lomb, Rochester, NY 14609, USA; heleen.decory@bausch.com

**Keywords:** besifloxacin, benzalkonium chloride, in vitro, time-kill studies, ophthalmic infections, antibacterial resistance

## Abstract

Background: Besifloxacin ophthalmic suspension 0.6% (*w*/*v*%) contains benzalkonium chloride (BAK) as a preservative. We evaluated the in vitro time-kill activity of besifloxacin, alone and in combination with BAK, against common bacteria implicated in ophthalmic infections. Methods: The activity of besifloxacin (100 µg/mL), BAK (10, 15, 20, and 100 µg/mL), and combinations of besifloxacin and BAK were evaluated against isolates of *Staphylococcus epidermidis* (*n* = 4), *Staphylococcus aureus* (*n* = 3), *Haemophilus influenzae* (*n* = 2), and *Pseudomonas aeruginosa* (*n* = 2) in time-kill experiments of 180 min duration. With the exception of one *S. aureus* isolate, all of the staphylococcal isolates were methicillin- and/or ciprofloxacin-resistant; one *P. aeruginosa* isolate was ciprofloxacin-resistant. The reductions in the viable colony counts (log_10_ CFU/mL) were plotted against time, and the differences among the time–kill curves were evaluated using an analysis of variance. Areas-under-the-killing-curve (AUKCs) were also computed. Results: Besifloxacin alone demonstrated ≥3-log killing of *P. aeruginosa* (<5 min) and *H. influenzae* (<120 min), and approached 3-log kills of *S. aureus*. BAK alone demonstrated concentration-dependent killing of *S. epidermidis*, *S. aureus* and *H. influenzae*, and at 100 µg/mL produced ≥3-log kills in <5 min against these species. The addition of BAK (10, 15, and 20 µg/mL) to besifloxacin increased the rate of killing compared to besifloxacin alone, with earlier 3-log kills of all species except *P. aeruginosa* and a variable impact on *S. aureus*. The greatest reductions in AUKC were observed among *H. influenzae* (8-fold) and *S. epidermidis* (≥5-fold). Similar results were found when the isolates were evaluated individually by their resistance phenotype. Conclusions: In addition to confirming the activity of 100 µg/mL BAK as a preservative in the bottle, these data suggest that BAK may help besifloxacin to achieve faster time-kills on-eye in the immediate timeframe post-instillation before extensive dilution against bacterial species implicated in ophthalmic infections, including drug-resistant *S. epidermidis*. Greater killing activity may help prevent resistance development and/or help treat resistant organisms.

## 1. Introduction

Fluoroquinolones are broad-spectrum, bactericidal antibiotics that inhibit DNA synthesis through dual actions on the bacterial enzymes DNA gyrase (topoisomerase II) and topoisomerase IV [1]. Besifloxacin is a chlorinated fluoroquinolone introduced into clinical practice in the US in 2009, and is available worldwide including in Canada, Australia, and several Latin American, Middle Eastern, African, and Asian countries. Unlike other fluoroquinolones, besifloxacin was developed and formulated solely for topical ocular administration and is indicated in the US for the treatment of bacterial conjunctivitis [2]. Studies have shown that the minimum inhibitory concentration (MIC) values for besifloxacin against Gram-positive pathogens are consistently lower, by at least two to three dilutions, than those for moxifloxacin and gatifloxacin against the same isolates [3,4,5], and that besifloxacin was also the most rapidly bactericidal [6]. Besifloxacin ophthalmic suspension 0.6% (Besivance^®^, Bausch + Lomb) has been studied clinically for the treatment of bacterial conjunctivitis [7,8,9,10,11,12,13,14] and other potential uses [15,16,17,18,19].

The marketed formulation of besifloxacin includes the drug at a concentration of 6 mg/mL (0.6%) and the preservative benzalkonium chloride (BAK) at a concentration of 100 µg/mL (0.01%) [2]. The quaternary ammonium compound BAK is the most widely used preservative in ophthalmic medications and has been shown to have antimicrobial activity of its own. BAK is a detergent preservative and interacts with lipid components of the bacterial cell membrane leading to membrane destabilization, release of cell contents, and ultimately cell death [20]. BAK has demonstrated significant in vitro antimicrobial activity against common Gram-positive cocci such as *Staphylococcus aureus* and *Staphylococcus epidermidis*, but much less so on Gram-negative rods such as *Pseudomonas aeruginosa* [21,22]. In vehicle-controlled clinical conjunctivitis studies, 0.6% besifloxacin ophthalmic suspension containing 0.01% BAK was statistically superior to vehicle alone with respect to both clinical and microbiological resolution at almost every follow up visit; however, a percentage of subjects using only vehicle, which contained BAK, also achieved successful outcomes [7,8,10]. While bacterial conjunctivitis is typically self-resolving given enough time [23], these findings suggest that BAK may contribute to microbial resolution in vehicle-treated subjects and may enhance microbial resolution with besifloxacin in the marketed formulation.

Our laboratory (JB) previously found MIC values of ≤3.1 µg/mL when BAK was tested against isolates of methicillin-susceptible *S. aureus* (MSSA) [24], methicillin-resistant *S. aureus* (MRSA) [24,25], and coagulase-negative staphylococci (CoNS) [24]. Additionally, when gatifloxacin or moxifloxacin was tested in vitro in combination with varying concentrations of BAK against these pathogens, the MIC values were multiple-fold lower (all ≤0.008 µg/mL) compared to the MIC values for the fluoroquinolone agents alone [24,25].

MIC data are useful in comparing the potency of antibacterial agents to one another against a specific pathogenic organism. However, given concerns about increasing bacterial resistance and mutant bacteria selection, the potential of a drug or combination of drugs to completely eradicate an organism, which is referred to as kill data, can provide information that is perhaps more relevant. In the current study, in vitro time-kill studies were performed to compare the activity of besifloxacin alone, BAK alone at increasing concentrations, and the combination of besifloxacin and BAK against common ophthalmic Gram-positive and Gram-negative pathogens implicated in ocular infections, including isolates with varying antibacterial resistance profiles.

## 2. Results

Figure 1 presents the log reductions in viable cell counts (CFU/mL) of *S. epidermidis*, *S. aureus*, *H. influenzae,* and *P. aeruginosa* following incubation with BAK alone over 180 min. BAK demonstrated concentration-dependent killing of *S. epidermidis*, *H. influenzae*, and *S. aureus*. At the highest concentration tested (100 µg/mL), BAK produced a >5-log kill within 5 min against *S. epidermidis*, *S. aureus*, and *H. influenzae*. As expected, BAK did not demonstrate robust killing activity against *P. aeruginosa*, although a ≥3-log kill was seen with the highest concentration of BAK, but only after 180 min incubation.

Figure 2 presents log reductions in viable counts for besifloxacin alone at 100 µg/mL and in combination with increasing concentrations of BAK (10 µg/mL, 15 µg/mL, 20 µg/mL) for each species (average of isolates thereof) tested over 180 min. The corresponding log reductions at the final timepoint are presented in Table 1. Results of the ANOVA of kill curves, excluding those with BAK at 100 µg/mL, indicated a significant effect of test agent (i.e., besifloxacin and/or BAK) on kill curves for *S. epidermidis*, *H. influenzae,* and *P. aeruginosa* (*p* ≤ 0.0092), as well as a significant effect of time and interaction of time with test agent for all four species (*p* ≤ 0.0157).

With 100 µg/mL besifloxacin alone, a 0.3-log reduction was seen in viable *S. epidermidis* organisms following 5 min of drug exposure, increasing to a 1.4-log reduction at 180 min of drug exposure (Figure 2). While the addition of BAK at 10 µg/mL provided no additional killing of *S. epidermidis* organisms, the addition of BAK at 15 µg/mL and BAK at 20 µg/mL increased the killing, resulting in 3-log reductions in viable *S. epidermidis* organisms by about 60 min for the combination of besifloxacin and 15 µg/mL BAK, and in less than 30 min for the combination of besifloxacin and 20 µg/mL BAK. The addition of BAK at 15 and at 20 µg/mL to besifloxacin significantly increased the bacterial killing of *S. epidermidis* compared to that with besifloxacin alone at the timepoints 60 min and 20 min onward, respectively (*p* ≤ 0.003). The corresponding AUKC for the combination of besifloxacin and BAK at 15 µg/mL and besifloxacin and 20 µg/mL BAK were decreased 2.5- and 5.6-fold from the AUKC for besifloxacin alone. Comparison of kill curves for BAK alone compared to besifloxacin alone showed a significant difference at the timepoints 60 min and 20 min onward for BAK at 15 and BAK at 20 µg/mL, respectively (*p* ≤ 0.049, data not shown), with greater activity for BAK at these concentrations vs besifloxacin alone at these timepoints.

Against *S. aureus*, 100 µg/mL besifloxacin alone produced a 0.6-log reduction in viable organisms after 5 min of drug exposure, increasing to a 2.6-log reduction after 180 min of drug exposure. While the addition of BAK at 10 µg/mL or at 15 µg/mL to besifloxacin had no effect on log reductions of viable *S. aureus* organisms compared to those observed with besifloxacin alone, the addition of BAK at 20 µg/mL to besifloxacin produced a 3-log reduction in viable *S. aureus* organisms by roughly 80 min and a significantly increased bacterial kill at the 180 min timepoint on the kill curve for the combination compared to besifloxacin alone (*p* = 0.0358). The corresponding AUKC for the combination was also decreased 1.6-fold from that with besifloxacin alone. Comparison of the kill curve of 100 µg/mL besifloxacin alone to kill curves of BAK alone showed a significant difference to that of BAK at 10 µg/mL at the 180 min timepoint (*p* = 0.0275; data not shown), with significantly greater activity of besifloxacin alone vs BAK at 10 µg/mL.

Against *H. influenzae,* besifloxacin alone at 100 µg/mL produced a 1.3-log reduction in viable counts by 5 min of exposure, increasing to a 3.0-log reduction by 120 min of exposure. The addition of BAK at 15 µg/mL or 20 µg/mL to besifloxacin resulted in 3-log reductions in viable *H. influenzae* organisms in just over 10 min for both combinations. Furthermore, the addition of 15 µg/mL BAK to besifloxacin produced significantly increased bacterial killing of *H. influenzae* at the 60-min timepoint (*p* = 0.0320), while the addition of 20 µg/mL BAK to besifloxacin produced significantly increased bacterial killing of *H. influenzae* at timepoints from 20 min to 120 min compared to besifloxacin alone (*p* ≤ 0.0412). The AUKC for besifloxacin combined with 15 µg/mL BAK or combined with 20 µg/mL BAK were decreased 4.4- and 8.2-fold from that with besifloxacin alone. Comparison of the kill curve of 100 µg/mL besifloxacin alone to the kill curves of BAK alone showed a significant difference to that of BAK at 20 µg/mL at the 60-min timepoint (*p* = 0.0024; data not shown), with greater killing activity for BAK.

A 3.2-log reduction in viable cell counts of *P. aeruginosa* was observed with 100 µg/mL besifloxacin as early as 5 min of exposure, increasing to a 5.5-log reduction at 180 min. No significant increases in the reduction of viable cell counts of *P. aeruginosa* were observed with the addition of BAK at any concentration to besifloxacin compared to besifloxacin alone at any timepoint. Likewise, the AUKCs for besifloxacin in combination with BAK at 10 µg/mL, 15 µg/mL, and 20 µg/mL were not reduced compared to that of besifloxacin alone. Notably, 100 µg/mL besifloxacin alone was significantly more active than BAK alone at 10 µg/mL, 15 µg/mL, and 20 µg/mL at all timepoints of the kill curve (*p* ≤ 0.0400; data not shown), and besifloxacin’s AUKC was reduced by approximately 9-fold from those with BAK alone at those concentrations.

Figure 3 presents the log reductions in viable cell counts over time for besifloxacin alone and in combination with increasing concentrations of BAK for *S. epidermidis* and *S. aureus* isolates categorized by their methicillin and/or ciprofloxacin resistance profile. Corresponding log reductions at the final timepoint are presented in Table 1. Results of the ANOVA for the kill curves for these individual isolates indicated a significant effect of test agent and time for these isolates regardless of the isolate resistance profile (*p* ≤ 0.0003 for all).

With besifloxacin alone, a 0.2-log kill was observed of the Cip^R^ MRSE isolate at 5 min drug exposure, which increased to a 0.4-log kill by 180 min, whereas more robust bacterial killing was observed of the two Cip^S^ MRSE isolates, with a 0.6-log kill observed at 5 min drug exposure, which increased to a 2.4-log kill by 180 min. The addition of BAK at 15 µg/mL or 20 µg/mL led to increased bacterial killing regardless of the ciprofloxacin resistance-phenotype. Thus, the combination of besifloxacin with BAK at 15 µg/mL or at 20 µg/mL, respectively, resulted in 3-log kills of the Cip^R^ MRSE isolate by about 90 and 45 min, and of the Cip^S^ MRSE isolates by about 50 and 30 min of combined drug exposure. Furthermore, the combination of BAK and besifloxacin was significantly more active than besifloxacin alone against the Cip^R^ MRSE isolate at the 180-min timepoint for BAK at 15 µg/mL; at the 60-min timepoint and onward for BAK at 20 µg/mL (*p* ≤ 0.0092); against the Cip^S^ MRSE isolates at the 60-min timepoints and onward for BAK at 15 µg/mL; and at the 20-min timepoint and onward for BAK at 20 µg/mL (*p* ≤ 0.0046). Minimal killing (<0.1-log kill) was observed with besifloxacin alone against the Cip^R^ MSSE isolate, increasing to 0.5 log units by 180 min. The addition of BAK to besifloxacin led to increased killing of this isolate, with 3-log kills achieved at approximately 80 min exposure for the combination of besifloxacin with 15 µg/mL BAK, and at 25 min for the combination of besifloxacin with 20 µg/mL BAK. Significant differences in the bacterial killing compared to besifloxacin alone were also found against this isolate at the 120-min timepoint and onward for BAK at 15 µg/mL, and at the 30-min timepoint and onward for BAK at 20 µg/mL (*p* ≤ 0.0110). The corresponding AUKCs for these *S. epidermidis* isolates were decreased by 2.2- to 2.7-fold with BAK at 15 µg/mL, and by 4.7- to 6.0-fold with BAK at 20 µg/mL from those with besifloxacin alone.

Besifloxacin alone at 100 µg/mL produced a minimal (<0.1 log) reduction in viable organisms of the Cip^R^ MRSA isolate after 5 min of exposure, increasing to a 1.2-log reduction by 180 min of drug exposure. Besifloxacin alone demonstrated greater activity against the MSSA isolates, with 1.0- and 0.9-log reductions at 5 min for the Cip^R^ and Cip^S^ MSSA isolates, respectively, increasing to 3-log reductions at 120 and 180 min. As was the case for the analysis of the kill curves for the three *S. aureus* isolates combined, the addition of BAK at 10 or 15 µg/mL to besifloxacin did not increase the log reductions in viable cells of the individual isolates at any timepoint. However, the addition of BAK at 20 µg/mL to besifloxacin increased the killing rate, with a 3-log reduction in viable cells observed at approximately 100 min for the Cip^R^ MRSA isolate, and by 20 min for the Cip^S^ MSSA isolate. A significantly increased bacterial kill was achieved for the combination of besifloxacin and BAK at 20 µg/mL compared to besifloxacin alone for the Cip^R^ MRSA isolate at 180 min (*p* = 0.0003) and the Cip^S^ MSSA isolate at 20, 25, and 30 min (*p* ≤ 0.0422). Notably, a reduction in the bacterial killing of the Cip^R^ MSSA isolate was observed when BAK at 10 or 15 µg/mL was added to besifloxacin, with significant differences observed at 120 min for the combination of besifloxacin and 10 µg/mL BAK, and at 180 min for both combinations compared to besifloxacin alone (*p* ≤ 0.0427). The AUKC for besifloxacin combined with 20 µg/mL BAK was decreased 1.7-fold for the Cip^R^ MRSA isolate and 3.7-fold for the Cip^S^ MSSA isolate from that with besifloxacin alone.

One of the *P. aeruginosa* isolates was ciprofloxacin-resistant, whereas the other was ciprofloxacin sensitive. The evaluation of kill curves for these individual isolates demonstrated a 1.0-log reduction in the viable counts of the Cip^R^
*P. aeruginosa* isolate, increasing to a 3-log reduction at 30 min, and a 5.4-log reduction in the viable counts of the Cip^S^
*P. aeruginosa* isolate already at 5 min of exposure. The addition of BAK to besifloxacin appeared to increase the bacterial killing rate of the Cip^R^
*P. aeruginosa* isolate slightly, helping to achieve 3-log reductions in viable cells by approximately 10 min through the addition of BAK at 15 µg/mL or at 20 µg/mL to besifloxacin, and by 20 min through the addition of BAK at 10 µg/mL to besifloxacin. However, no statistically significant increases were noted over besifloxacin alone at any of the added BAK concentrations. Likewise, the AUKCs for besifloxacin in combination with BAK at 10 µg/mL, 15 µg/mL, and 20 µg/mL were not reduced compared to that of besifloxacin alone.

## 3. Discussion

Bacterial resistance to antibacterial agents and/or the development of resistance is an ongoing concern in all fields of medicine, including ophthalmology. While MICs are useful in the evaluation of the relative potencies of antibiotics, an MIC only measures the degree of growth inhibition at a given drug concentration. Despite a low MIC and an organism labeled as “susceptible,” it is possible for mutant organisms to survive and/or develop, especially when the bacterial load is large. In vitro bacterial time-kill experiments, however, provide data regarding the rapidity and extent of killing by antibiotics, both of which may be important clinically, and are likely to be more relevant in the context of understanding the potential or lack of potential for resistance development. Indeed, as famously expressed by Charles Nightingale (Hartford Hospital, Hartford, CT) and subsequently adapted by others [26,27] “dead bugs don’t mutate.”

In this in vitro time-kill study, BAK alone demonstrated concentration-dependent killing of three of the four common ocular bacterial pathogens tested. For *S. epidermidis*, *H. influenzae* and *S. aureus*, the rate and extent of the organism killing were greater with increasing concentrations of BAK, and at the highest BAK concentration tested (100 µg/mL), these organisms were killed completely within the first 5 min of drug exposure. However, as is consistent with prior reports [28,29,30], BAK alone had poor activity against *P. aeruginosa*. In contrast, besifloxacin alone at 100 µg/mL demonstrated killing of all four of these common pathogenic species, including *P. aeruginosa*. Besifloxacin’s bacterial killing was greatest against *P. aeruginosa*, despite one of the isolates tested being ciprofloxacin-resistant, with a mean log kill of >3 after only 5 min of exposure, increasing to 5.5-log kills (the maximum possible) by 120 min, followed by *H. influenzae*, with a mean log kill of 1.3 after only 5 min of drug exposure, increasing to 3.6-log reductions after 180 min. Bacterial killing was also observed against *S. epidermidis* and *S. aureus,* with mean log reductions of 0.3 and 0.6, respectively, after 5 min, increasing to log kills of 1.4 and 2.6 at 180 min.

The objective of this study was to evaluate the effect of added BAK on the in vitro bacterial killing of besifloxacin. Against *S. epidermidis* and *H. influenzae* isolates, the addition of BAK to besifloxacin increased the rate of bacterial killing compared to besifloxacin alone in proportion to the concentration of the BAK added, whereas the addition of BAK had a variable impact against *S. aureus* in which increased bacterial killing was only found with the addition of 20 µg/mL BAK, and only in two of the three isolates tested. In general, 3-log kills were achieved faster through the addition of BAK to besifloxacin compared to besifloxacin alone, and statistically significant increased killing was even found at specific timepoints in the time–kill curves, particularly for the concentration of BAK at 20 µg/mL in combination with besifloxacin compared to besifloxacin alone. All of the *S. epidermidis* isolates tested in this study were methicillin- and/or ciprofloxacin-resistant, and these findings were likewise observed for *S. epidermidis* strains when kill curves were analyzed according to the isolate(s) resistance profile. Corresponding reductions in the AUKC were also observed with the addition of BAK to besifloxacin in most instances, again signifying greater and/or faster killing with the combination of besifloxacin and BAK of these three common pathogenic species. However, as expected, there was no significant impact of BAK on the killing effect of besifloxacin against *P. aeruginosa*, despite the appearance of slightly increased killing of the Cip^R^ isolate.

Our findings expand on prior studies demonstrating increased killing effects when relatively high concentrations of BAK are added to a fluoroquinolone. Haas et al. [6] performed in vitro time-kill experiments evaluating besifloxacin, moxifloxacin, and gatifloxacin against *S. aureus*, *S. epidermidis,* and *H. influenzae*, with and without BAK. Regardless of the bacterial species, viable cell counts were reduced to the lowest level of detection within the first 5 min when evaluating the fluoroquinolones combined with BAK at 50 or 100 µg/mL. Consistent with our results, concentration-dependent killing was observed at BAK concentrations <50 µg/mL. However, the authors failed to observe increased killing for BAK at concentrations <50 µg/mL in combination with besifloxacin, likely due to the concentration of besifloxacin tested, namely 0.24 µg/mL, which was much lower than that tested in our study [6]. In another in vitro study evaluating the kill rates of commercial formulations, 0.3% gatifloxacin ophthalmic suspension (formulated with BAK 50 µg/mL) was found to eradicate *S. aureus* and CoNS more rapidly than the 0.5% moxifloxacin ophthalmic solution, which does not contain BAK [31].

Clearly, the data presented here are in vitro findings, and the time-kill data reflect the continued exposure of the bacterial inoculum to a particular concentration of besifloxacin alone or in combination with different concentrations of BAK for several hours, whereas with clinical administration to the ocular surface, natural physiologic processes lead to a rapid dilution of both besifloxacin and BAK in the tear film. In the current in vitro study, the concentration of besifloxacin tested, namely 100 µg/mL, is 60-fold lower than that in the instilled drop (0.6% or 6 mg/mL). Proksch et al. reported a besifloxacin maximum tear concentration of 610 µg/g at 10 min following a single drop administration to healthy volunteers, which decreased to approximately 50 μg/g at 8 h, 10 μg/g at 12 h, and still more than 1 μg/g at 24 h [32]. Thus, a 100 µg/mL concentration of besifloxacin is a reasonable estimate of the on-eye concentration of besifloxacin that is clinically achievable and sustained for a significant time following instillation of a drop; the in vitro findings herein for besifloxacin alone are therefore reassuring and confirm the expected clinical activity of besifloxacin. In contrast, the highest concentration of BAK tested in vitro, also 100 µg/mL, is the same as that in the bottle, and represents undiluted BAK. It follows that the log reduction findings for the 100 µg/mL BAK concentration are relevant only to bactericidal activity in the bottle and are supportive of the intended use of BAK as a preservative in the bottle. However, the findings for BAK at the lower concentrations of 10 µg/mL, 15 µg/mL, and 20 µg/mL, representing 10-, 6.7-, and 5-fold dilutions, respectively, may roughly represent the concentrations of BAK achieved on-eye, especially in the immediate time frame after the drop instillation. The testing of BAK at lower concentrations was not reasonable given the reported MIC of BAK of ≤3.1 µg/mL against strains of methicillin-sensitive and -resistant *S. aureus* and CoNS, along with reporting by Friedlaender et al. of a 16-fold dilution of BAK in tears (to only 3.2 µg/mL) as early as 1 min after the instillation of another fluoroquinolone formulation containing 0.005% (50 µg/mL) BAK [33].

However, even as the BAK concentrations are diluted rapidly during the first few minutes after instillation, the expected increased killing effects observed with BAK in combination with besifloxacin in this immediate time frame post-instillation compared to besifloxacin alone could very well be clinically meaningful, especially against *S. epidermidis*, some *S. aureus*, and *H. influenzae* organisms, and may help minimize the potential for microbial resistance development in these species, given that the greater and/or faster the killing of a pathogenic bacteria is, the smaller the risk of resistance development. Moreover, an increased rate of kill may also help mitigate infection with isolates which are already resistant. While little antibacterial resistance has been reported among *H. influenzae* in ocular infections to date, in vitro resistance has been reported among CoNS and *S. aureus* [5,34,35,36,37], and MRSA has become a growing concern in ophthalmic infections [38,39,40,41,42,43]. In this context, it is notable that the addition of BAK at 20 µg/mL to besifloxacin was especially impactful with regards to the achievement of 3-log kills of Cip^R^ and Cip^S^ MRSE as well as Cip^R^ MSSE by approximately 30 min. Similarly, the addition of BAK at 20 µg/mL to besifloxacin decreased the time of exposure needed to achieve the 3-log killing of the Cip^R^ MRSA isolate compared to besifloxacin alone. However, the addition of BAK did not increase the bacterial killing rate of the one Cip^R^ methicillin-sensitive *S. aureus* isolate tested in this study. In fact, BAK appeared to paradoxically decrease the bacterial kill rate of that isolate when added at concentrations of either 10 and 15 µg/mL. The reason for this finding is unclear and warrants further investigation.

The limitations of this study include the lack of timepoints between 0 and 5 min of drug exposure, the small number of isolates tested, and the inability of time-kill experiments to accurately reflect the changing concentrations of either besifloxacin and BAK on-eye following the instillation of a drop of besifloxacin ophthalmic suspension 0.6%. However, in general, the findings of this in vitro study do suggest that the inclusion of BAK at 0.01% in the besifloxacin formulation could enhance the bacterial killing with besifloxacin in the immediate period post-instillation on-eye, in particular that of *S. epidermidis* and *S. aureus* with varying resistance profiles, and that of *H. influenzae*, thereby aiding in the prevention of antibiotic resistance development and/or the treatment of already drug-resistant isolates. Finally, the MICs of besifloxacin for the isolates tested were not known, nor would they have been interpretable given the current absence of established CLSI breakpoints for besifloxacin. Nevertheless, microbiological studies to date indicate that the MIC of besifloxacin is normally within one to two dilutions of that of moxifloxacin against methicillin-sensitive *S. aureus* and methicillin-sensitive CoNS, and three dilutions lower than that of moxifloxacin against methicillin-resistant staphylococci [5]. Thus, the concentration of besifloxacin tested in this study, 100 µg/mL, is likely to be several-fold higher than the 4X MIC typically used as the lower limit in time-kill studies. More importantly, the concentration studied was selected based on the known/expected concentrations of besifloxacin on-eye following the instillation of 0.6% besifloxacin ophthalmic solution.

In conclusion, the addition of BAK to besifloxacin led to rapid eradication of common ocular pathogens under in vitro conditions. While the clinical significance of enhanced antibacterial killing with the addition of BAK cannot be determined with certainty, these data support the inclusion of BAK in the besifloxacin formulation and in other topical ophthalmic fluoroquinolone formulations. Beyond its use as a preservative in the bottle, BAK may contribute to the killing of bacterial pathogens immediately post-instillation and thereby help to suppress antibiotic resistance development and/or aid in the eradication of bacterial strains that are already resistant.

## 4. Methods

Clinical, non-duplicate ocular isolates of four bacterial species were collected at the Clinical Microbiology Laboratory of the Royal University Hospital, Saskatoon, Saskatchewan, Canada, including *S. epidermidis* (4 isolates), *S. aureus* (3 isolates), *P. aeruginosa* (2 isolates), and *Haemophilus influenzae* (2 isolates). The staphylococcal isolates had varying drug resistance phenotypes: three of the *S. epidermidis* isolates and one *S. aureus* isolate were oxacillin-resistant (i.e., methicillin-resistant; MRSE and MRSA, respectively), and two *S. epidermidis* and two *S. aureus* isolates (including the oxacillin-resistant one) were ciprofloxacin-resistant (Cip^R^). The respective MICs for ciprofloxacin and moxifloxacin against staphylococcal isolates were 64 and 16 µg/mL for Cip^R^ MRSE, ≤0.063 and ≤0.031 µg/mL for ciprofloxacin-sensitive (Cip^S^) MRSE, 32 and 16 µg/mL for Cip^R^ MSSE, ≥64 and ≥16 µg/mL for Cip^R^ MRSA, 8 and 0.25 µg/mL Cip^R^ MSSA, and 0.125 and 0.031 µg/mL for Cip^S^ MSSA. Additionally, one *P. aeruginosa* isolate was ciprofloxacin resistant.

Institutional review board approval was not required as this was a laboratory study. The isolates were obtained from specimens submitted during standard care, and no patient-specific information was collected or recorded for these isolates for the purposes of the study.

Besifloxacin and BAK were obtained from Bausch & Lomb Incorporated (Rochester, NY). Methods for culture preparation and time-kill studies were described previously [44,45,46]. Briefly, *S. epidermidis*, *P. aeruginosa,* and *S. aureus* isolates were grown overnight on Tryptic Soy Agar (TSA) plates containing 5% sheep blood, whereas the *H. influenzae* isolates were grown on TSA plates containing 5% defibrinated sheep blood (chocolate plates). On the following day, an inoculum of each isolate was transferred to test tubes containing growth media appropriate to the species (i.e., *S. epidermidis*, *S. aureus,* and *P. aeruginosa* were transferred to Mueller–Hinton Broth and *H. influenzae* isolates were transferred to Haemophilus Test Media) and incubated for 2 h at 35–37 °C (in ambient air for staphylococci and *P. aeruginosa,* or in 5% CO_2_ for *H. influenzae*) in order to ensure that the bacterial isolates were in the growth phase.

For the time-kill experiments, isolate cultures were evaluated spectrophotometrically and diluted as necessary to achieve final cell densities of 10^4^–10^5^ cells/mL, and antimicrobial agents were added to the culture tubes to achieve final concentrations of 10, 15, 20, or 100 µg/mL BAK, 100 µg/mL besifloxacin, or combinations of besifloxacin (100 µg/mL) and BAK (10, 15, 20, or 100 µg/mL). Following 5, 10, 15, 20, 25, 30, 60, 120, and 180 min of exposure, triplicate sample aliquots were removed, serially diluted, and plated on appropriate drug-free agar medium. After incubation overnight at 35–37 °C (in 5% CO_2_ or ambient air, based on the species) viable colonies were enumerated.

The rate and extent of the bacterial killing were determined by plotting the reduction in viable colony counts (log_10_ CFU/mL) against time. Differences in time-kill curves were evaluated using two-way analysis of variance (ANOVA) followed by the Dunnett’s multiple comparisons test. A *p* value of <0.05 was considered significant. Additionally, areas-under-the-killing-curve (AUKC) over 180 min were computed using the trapezoid rule and a fixed baseline of −5.2 log units, and fold-reductions in the AUKC were calculated. Data were plotted and analyzed using GraphPad Prism version 6.07 for Windows (GraphPad Software, La Jolla, CA, USA).

## Figures and Tables

**Figure 1 pharmaceuticals-14-00517-f001:**
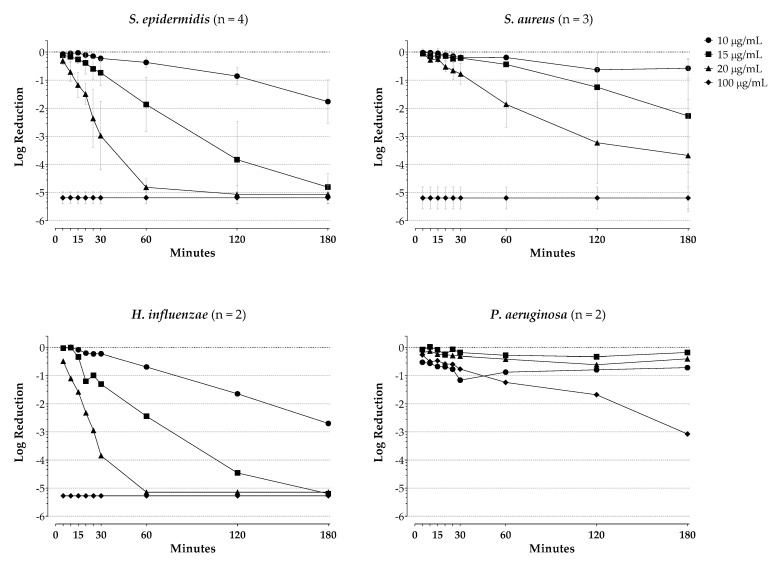
Log reductions in viable cells over time with increasing concentrations of BAK alone. Starting (0 min) viable cell densities for each isolate were ~10^5^ CFU/mL. Each plot point represents the mean (±SD) (*S. epidermidis* and *S. aureus*) or average (*H. influenzae* and *P. aeruginosa*) for the isolates tested of each species.

**Figure 2 pharmaceuticals-14-00517-f002:**
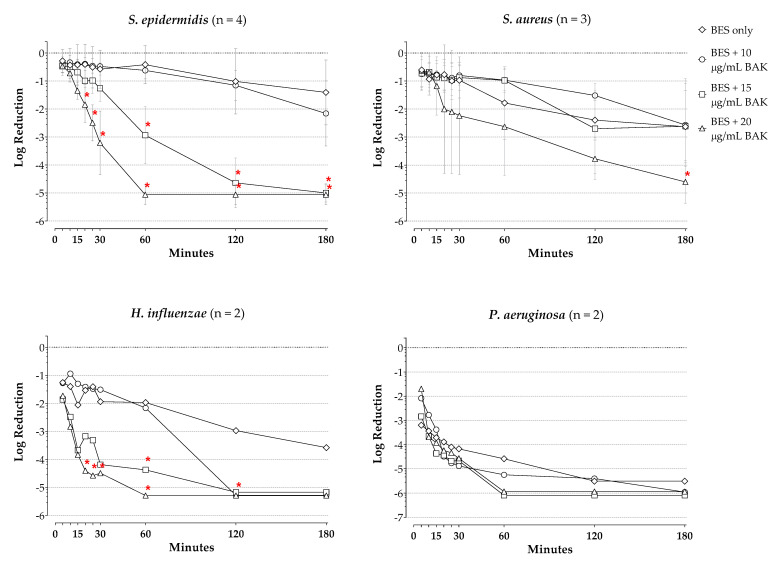
Log reductions in viable cells over time with 100 µg/mL besifloxacin alone or combined with increasing amounts of BAK. Starting (0 min) viable cell densities for each isolate were ~10^5^ CFU/mL. Each plot point represents the mean (±SD) (*S. epidermidis* and *S. aureus*) or average (*H. influenzae* and *P. aeruginosa*) for the isolates tested of each species. * *p* < 0.05 compared with besifloxacin. BES = besifloxacin, BAK = benzalkonium chloride.

**Figure 3 pharmaceuticals-14-00517-f003:**
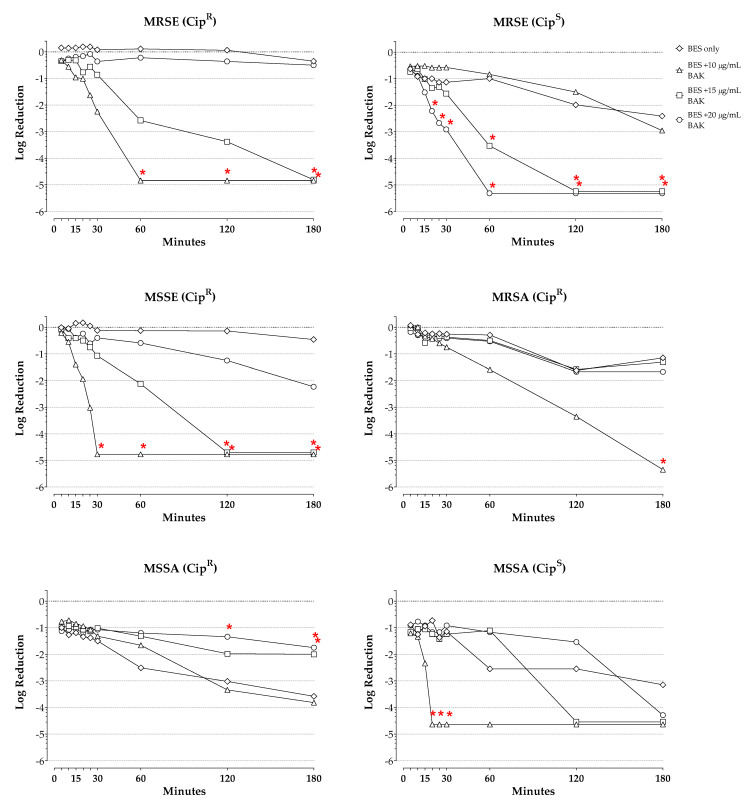
Log reductions in viable organisms over time with 100 µg/mL besifloxacin alone or combined with increasing amounts of BAK for *S. epidermidis* and *S. aureus* isolates by resistance profile. Starting (0 min) viable cell densities for each isolate were ~10^5^ CFU/mL. All of the plots represent one tested isolate, with the exception of MRSE (Cip^S^), which represents the average of two tested isolates. * *p* < 0.05 compared with besifloxacin. BES = besifloxacin, BAK = benzalkonium chloride, MRSE = methicillin-resistant *S. epidermidis*, MSSE = methicillin-sensitive *S. epidermidis*, MRSA = methicillin-resistant *S. aureus*, MSSA = methicillin-sensitive *S. aureus*, Cip^R^ = ciprofloxacin-resistant, Cip^S^ = ciprofloxacin-sensitive.

**Table 1 pharmaceuticals-14-00517-t001:** Bacterial kill after 180 min for besifloxacin alone and besifloxacin in combination with BAK.

	BES (100 µg/mL)	BES ^a^ + BAK (10 µg/mL)	BES ^a^ + BAK (15 µg/mL)	BES ^a^ + BAK (20 µg/mL)
***S. epidermidis***, *n* = 4				
Mean Δlog CFU/mL	−1.41	−2.16	−5.00	−5.06
Mean Difference (95% CI)		0.753 (−0.306, 1.81)	3.59 (2.53, 4.65)	3.66 (2.60, 4.71)
*p*-value ^b^		NS	<0.0001	<0.0001
**Cip^R^ MRSE**, *n* = 1				
Δlog CFU/mL	−0.350	−0.500	−4.81	−4.81
Mean Difference (95% CI)		0.150 (−3.43, 3.73)	4.46 (0.876, 8.04)	4.49 (0.906, 8.07)
*p*-value ^b^		NS	0.0092	0.0086
**Cip^S^ MRSE**, *n* = 2				
Mean Δlog CFU/mL	−2.41	−2.96	−5.24	−5.32
Mean Difference (95% CI)		0.545 (−0.463, 1.55)	2.83 (1.82, 3.84)	2.91 (1.90, 3.92)
*p*-value ^b^		NS	<0.0001	<0.0001
**Cip^R^ MSSE**, *n* = 1				
Δlog CFU/mL	−0.460	−2.23	−4.70	−4.77
Mean Difference (95% CI)		1.77 (−1.71, 5.25)	4.24 (0.765, 7.72)	4.31 (0.835, 7.79)
*p*-value ^b^		NS	0.0110	0.0095
***S. aureus***, *n* = 3				
Mean Δlog CFU/mL	−2.63	−2.57	−2.62	−4.60
Mean Difference (95% CI)		−0.0567 (−1.81, 1.70)	−0.0100 (−1.77, 1.75)	1.98 (0.221, 3.73)
*p*-value ^b^		NS	NS	0.0196
**Cip^R^ MRSA**, *n* = 1				
Δlog CFU/mL	−1.15	−1.67	−1.31	−5.35
Mean Difference (95% CI)		0.520 (−1.96, 3.00)	0.160 (−2.32, 2.64)	4.20 (1.72, 6.68)
*p*-value ^b^		NS	NS	0.0003
**Cip^R^ MSSA**, *n* = 1				
Δlog CFU/mL	−3.58	−1.75	−2.00	−3.82
Mean Difference (95% CI)		−1.83 (−3.37, −0.29)	−1.58 (−3.12, −0.04)	0.024 (−1.30, 1.78)
*p*-value ^b^		0.0141	0.0427	NS
**Cip^S^ MSSA**, *n* = 1				
Δlog CFU/mL	−3.15	−4.29	−4.54	−4.64
Mean Difference (95% CI)		1.14 (−2.05, 4.33)	1.39 (−1.80, 4.58)	1.49 (−1.70, 4.68)
*p*-value ^b^		NS	NS	NS
***H. influenzae***, *n* = 2				
Mean Δlog CFU/mL	−3.58	−5.29	−5.18	−5.30
Mean Difference (95% CI)		1.71 (−0.380, 3.79)	1.60 (−0.490, 3.68)	1.72 (−0.370, 3.80)
*p*-value ^b^		NS	NS	NS
***P. aeruginosa***, *n* = 2				
Mean Δlog CFU/mL	−5.51	−5.96	−6.10	−5.94
Mean Difference (95% CI)		0.450 (−2.27, 3.17)	0.585 (−2.13, 3.30)	0.430 (−2.29, 3.15)
*p*-value ^b^		NS	NS	NS
**Cip^R^*P. aeruginosa***, *n* = 1				
Δlog CFU/mL	−5.63	−5.93	−5.99	−5.97
Mean Difference (95% CI)		0.300 (−2.86, 3.46)	0.360 (−2.80, 3.52)	0.340 (−2.82, 3.50)
*p*-value ^b^		NS	NS	NS
**Cip^S^*P. aeruginosa***, *n* = 1				
Δlog CFU/mL	−5.39	−5.99	−6.20	−5.91
Mean Difference (95% CI)		0.600 (−1.77, 2.97)	0.810 (−1.56, 3.18)	0.520 (−1.85, 2.89)
*p*-value ^b^		NS	NS	NS

BES, besifloxacin; BAK, benzalkonium chloride; CFU/mL, colony forming units per mL; CI, confidence interval; NS, not significant; Cip^R^, ciprofloxacin-resistant; Cip^S^, ciprofloxacin sensitive; MRSE, methicillin-resistant *S. epidermidis*; MSSE, methicillin-sensitive *S. epidermidis*; MRSA, methicillin-resistant *S. aureus*; MSSA, methicillin-sensitive *S. aureus*. ^a^ 100 µg/mL. ^b^
*p* value from Dunnett’s multiple comparisons test. The respective MICs for ciprofloxacin and moxifloxacin against staphylococcal isolates were 64 and 16 µg/mL for Cip^R^ MRSE, ≤0.063 and ≤0.031 µg/mL for Cip^S^ MRSE, 32 and 16 µg/mL for Cip^R^ MSSE, ≥64 and ≥16 µg/mL for Cip^R^ MRSA, 8 and 0.25 µg/mL Cip^R^ MSSA, and 0.125 and 0.031 µg/mL for Cip^S^ MSSA.

## Data Availability

The data presented in this study are available in this article, “In Vitro Time-Kill of Common Ocular Pathogens with Besifloxacin Alone and in Combination with Benzalkonium Chloride”.

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
