# Peer review of "In Vitro Time-Kill of Common Ocular Pathogens with Besifloxacin Alone and in Combination with Benzalkonium Chloride"

_pharmaceuticals, 2021, doi:10.3390/ph14060517_

Round 1

Reviewer 1 Report

The manuscript presented by Blondeau and DeCory describes the usage of besifloxacin and its combination with benzalkonium chloride against common ocular bacteria. Authors tested clinical strains of both Gram+ (S. aureus, S. epidermidis) and Gram- (H. influenza, P. aeruginosa) bacteria which I found very accurate in such study. In general, I think this is smooth elaboration on the scientific problem however some data are missing and should be improved. 

Some general comments: 

  • Figures 1,2,3, and 4 - Do not show time-points at 0 minutes (although mentioned in the discussion as limitations). Please explain or include those time points.
  • Why did authors perform experiments only up to 180 minutes and not up to 24h? Would we see even more reduction in viability if experiments were prolongated? 
  • Could authors include the exact values of CFU/ml for figures 1,2,3, and 4 not only log reduction? It'd be much more informative. I suggest either to do it in SI or to change values of log reduction to exact CFU/ml. 
  • In some panels we see a plateau effect after treatment with BAK (see e.g. S. epidermidis 60-180 minutes with 20 ug/ml BAK or H. influenza (same scenario) however without values of CFU/ml we cannot asses if it was complete eradication of bacteria or it is a classical persisters plateau. 
  • We need to see figures presenting reduction in CFU/mL (or log reduction) for BES in various concentrations as well. This is unclear why authors decided to use only 100 µg/mL of BES and we don't know what is the effect of BES 100ug/ml alone on tested strains. 
  • I haven't found Table1 informative or easy to read. Also in the text, there is poor reference and description of the Table 1. Please, think again if this data is a must in the main text and if it is could it be presented in more readable way? Log reductions in CFU/mL for BES is a must to be able to asses values in Table 1.
  • Fractional inhibitory concentration (FIC) experiments for BES + BAK should be performed to estimate the effect and interaction between two drugs used in combination.
  • Quality of figure 3 is very poor and unacceptable.
  • Please correct some minor spelling and typos 

Author Response

Response to Reviewer 1 Comments

Point 1: The manuscript presented by Blondeau and DeCory describes the usage of besifloxacin and its combination with benzalkonium chloride against common ocular bacteria. Authors tested clinical strains of both Gram+ (S. aureus, S. epidermidis) and Gram- (H. influenza, P. aeruginosa) bacteria which I found very accurate in such study. In general, I think this is smooth elaboration on the scientific problem however some data are missing and should be improved.

Response 1: Thank you for your kind words.

Point 2: Figures 1,2,3, and 4 - Do not show time-points at 0 minutes (although mentioned in the discussion as limitations). Please explain or include those time points.

Response 2: The figures present log reductions in viable cells over time.  A sample was not taken at Time 0, or the start of the incubation, hence data points are not included in the figures for Time 0.  However, starting inoculums were determined at the beginning of the experiments and diluted appropriately to achieve 104 to 105 CFU/mL. Note that time-kill curves published in the medical literature conventionally start with the first sampling time point.

No changes made.

To clarify, in the current Discussion we mention that a limitation is that earlier time points were not included in our bacterial killing assessment (which in practice is very difficult to execute). By earlier time points we are referring to 1 minute, 2 minutes, 3 minutes of incubation etc.

Point 3: Why did authors perform experiments only up to 180 minutes and not up to 24h? Would we see even more reduction in viability if experiments were prolongated? 

Response 3: Yes, there would have been an even greater reduction in viable cells if time-kill experiments were prolonged beyond 3 hours for those isolates that were not killed completely by 180 minutes.   However, in this study we were interested in assessing bacterial killing with and without BAK in the shorter time frame to better model expected ocular pharmacokinetics on eye, given that antibiotics and BAK instilled on the eye are quickly diluted by tears. 

No change made.

Point 4: Could authors include the exact values of CFU/ml for figures 1,2,3, and 4 not only log reduction? It'd be much more informative. I suggest either to do it in SI or to change values of log reduction to exact CFU/ml.

Response 4: We initially attempted to do this, but because the starting CFUs/mL at 0 minutes varied between isolates and experiments, presenting log reductions in viable organisms with besifloxacin with and without BAK enabled us to better discern the impact of BAK on killing with besifloxacin.  No change made.

Point 5: In some panels we see a plateau effect after treatment with BAK (see e.g. S. epidermidis 60-180 minutes with 20 ug/ml BAK or H. influenza (same scenario) however without values of CFU/ml we cannot asses if it was complete eradication of bacteria or it is a classical persisters plateau.

Response 5: As mentioned in the methods, the starting concentration of viable cells was usually 104-105 cells/mL, hence there can be no more bacterial killing once you reach 5 log reductions as there are no more viable organisms to kill. 

We have added this information to the figure legends. 

Point 6: We need to see figures presenting reduction in CFU/mL (or log reduction) for BES in various concentrations as well. This is unclear why authors decided to use only 100 µg/mL of BES and we don't know what is the effect of BES 100 ug/ml alone on tested strains.

Response 6: Figures currently show log reductions.

Figure 2 and 3 do include besifloxacin at 100 µg/mL alone without BAK.  Please see the kill curves utilizing the diamond symbols therein. Note that we modified the figure legends to better convey that the kill curves utilizing the diamond symbols are for besifloxacin at 100 µg/mL alone by renaming the corresponding legend “BES only”.  

We did not test other concentrations of besifloxacin as we wanted to evaluate a concentration of the antibiotic that may be reasonably expected to be found on eye following topical instillation of besifloxacin ophthalmic solution 0.6%, as discussed in the Discussion of the paper.  See lines 333 to 340.  No change made.

Point 7: I haven't found Table1 informative or easy to read. Also in the text, there is poor reference and description of the Table 1. Please, think again if this data is a must in the main text and if it is could it be presented in more readable way? Log reductions in CFU/mL for BES is a must to be able to asses values in Table 1.

Response 7: We agree, and we assume that whilst inserting the table into the journal mandated template, the table became a little more difficult to read than our original table.   Note that log reduction achieved by 180 minutes with besifloxacin alone at 100 ug/mL is shown in the very first data column.

Please see reformatted table. We hope it is easier to follow and we ask the editor permission to retain the table in its current adjusted format for better comprehension.

Point 8: Fractional inhibitory concentration (FIC) experiments for BES + BAK should be performed to estimate the effect and interaction between two drugs used in combination.

Response 8: We thank the reviewer for his/her recommendation and will consider this assessment for future microbiological experiments. Note that our paper evaluated time kill curves and was not focused on MICs of the combination of besifloxacin and BAK.  However, we will consider the determination of FICs in the future.

Point 9: Quality of figure 3 is very poor and unacceptable.

Response 9: We agree that the figure quality is not acceptable as is.  We assume that the journal editorial staff will be able improve the quality of our figures in the ultimate proof (if accepted) as we will be providing the journal with original figure files as well.  Meanwhile, we have inserted our Figures using a different export format into the current revised version, and we feel doing so has resulted in a better image for purposes of viewing by reviewers.

Point 10: Please correct some minor spelling and typos

Response 10: We have reviewed the manuscript for additional spelling errors and typos and have fixed these (see tracked changes).

Reviewer 2 Report

In this manuscript, the authors evaluated the in vitro time-kill activity of besifloxacin, alone and in combination with BAK, against common bacteria implicated in ophthalmic infections. The manuscript is well written and the conclusions are supported by the data presented. Therefore, I only have one major comment as described below,

Major comment:

1. The authors evaluated the activity of besifloxacin (100 μg/mL), BAK (10, 15, 20, and 100 μg/mL), and combinations of besifloxacin plus BAK against ciprofloxacin resistant S. epidermidis, S. aureus, and P. aeruginosa. They should also evaluate the activity of ciprofloxacin (100 μg/mL) and combinations of ciprofloxacin plus BAK against these ciprofloxacin resistant isolates to clarify whether other fluoroquinolones in combination with besifloxacin (100 μg/mL) show high efficiency against ciprofloxacin resistant isolates.

Minor comments:

  1. line 109, "104–105 cells/mL" should be "104–105 cells/mL".
  2. lines 107 and 114, "CO2" should be "CO2"

Author Response

Response to Reviewer 2 Comments

Point 1: In this manuscript, the authors evaluated the in vitro time-kill activity of besifloxacin, alone and in combination with BAK, against common bacteria implicated in ophthalmic infections. The manuscript is well written and the conclusions are supported by the data presented. Therefore, I only have one major comment as described below,

Response 1: Thank you for your complimentary words.

Point 2: The authors evaluated the activity of besifloxacin (100 μg/mL), BAK (10, 15, 20, and 100 μg/mL), and combinations of besifloxacin plus BAK against ciprofloxacin resistant S. epidermidis, S. aureus, and P. aeruginosa. They should also evaluate the activity of ciprofloxacin (100 μg/mL) and combinations of ciprofloxacin plus BAK against these ciprofloxacin resistant isolates to clarify whether other fluoroquinolones in combination with besifloxacin (100 μg/mL) show high efficiency against ciprofloxacin resistant isolates.

Response 2: T The objective of our study was to evaluate the contribution of BAK to the killing activity of besifloxacin specifically. 

We agree it would be interesting to do similar studies with ciprofloxacin with and without BAK and assess differences to the current results with besifloxacin.  We may conduct such experiments in future.

Point 3: Minor comments:

  • line 109, "104–105 cells/mL" should be "104–105 cells/mL".
  • lines 107 and 114, "CO2" should be "CO2"

Response 3: We apologize for these formatting issues that resulted from inserting our manuscript into the journal template.  All subscripts and superscripts have been fixed.

Reviewer 3 Report

The manuscript entitled “In Vitro Time-Kill of Common Ocular Pathogens with Besifloxacin Alone and in Combination with Benzalkonium Chloride” is designed well. But the novelty of the manuscript is not significant. Authors address some of the edits before accepted for publication.

The title of the manuscript ‘in vitro time-kill’ – what is the exact definition of the phrase.

Previously, Elmer et al., reported the mox and gati with BAK activity.

Same company also published 2010 similar report. The novelty of the work is not significant.

Haas W, Pillar CM, Hesje CK, Sanfilippo CM, Morris TW. In vitro time–kill experiments with besifloxacin, moxifloxacin and gatifloxacin in the absence and presence of benzalkonium chloride. Journal of antimicrobial chemotherapy. 2011 Apr 1;66(4):840-4.

Hesje CK, Borsos SD, Blondeau JM. Benzalkonium chloride enhances antibacterial activity of gatifloxacin and reduces its propensity to select for fluoroquinolone-resistant strains. Journal of ocular pharmacology and therapeutics. 2009 Aug 1;25(4):329-34.

In abstract, line #9 write the concentration units of Bas suspension. What is the method used for the activity studies?

What is the level of significance getting for the data through ANOVA? Write the details in the results section of abstract.

In line #38, write the generation of Bas as fluoroquinolone.

Write the MIC values of Bas, Mox and gati against Gram + pathogens.

In case of BAK, write the details on BAK as permeation enhancer along with antimicrobial property.

Line #69, 76, in vitro should be in italics.

Line #103, an inoculum of each isolate – what is the expected colony count of each organism from this inoculum. How to check the consistency of the inoculum usage?

Line #106, 2 hours at 35–37oC, line#107, CO2  – units should be superscript and subscript.

Line#109, cell densities of 104–105 cells/mL – is it too the power of 4 – 5 or 104 – 105 cells/mL.

What is the reason for selection of testing concentrations of BAK and Bas for the study?

Line#119, P value of <0.05 – p should be small.

In Fig.1, include SD values and the clarity of the figure is very poor.

Why 100 ug/mL of BAK in combination with Bas missing in Figure 2.

It is better to split the Table 1 into organism wise. That will useful for the readers for easy comparison.

Author Response

Response to Reviewer 3 Comments

Point 1: The manuscript entitled “In Vitro Time-Kill of Common Ocular Pathogens with Besifloxacin Alone and in Combination with Benzalkonium Chloride” is designed well. But the novelty of the manuscript is not significant. Authors address some of the edits before accepted for publication.

Response 1: In our opinion the novelty lies in the application of the in vitro data to the in vivo clinical situation. There are few papers that discuss in vitro bacterial killing and how that may or may not be relevant to topically applied ocular antibiotics. 

Point 2: The title of the manuscript ‘in vitro time-kill’ – what is the exact definition of the phrase.

Response 2: We are unsure of the reviewer’s question.  This paper describes in vitro experiments evaluating bacterial killing over 180 minutes.  Please let us know if we have misunderstood the reviewers’ question.

Point 3: Previously, Elmer et al., reported the mox and gati with BAK activity.

Same company also published 2010 similar report. The novelty of the work is not significant.

Haas W, Pillar CM, Hesje CK, Sanfilippo CM, Morris TW. In vitro time–kill experiments with besifloxacin, moxifloxacin and gatifloxacin in the absence and presence of benzalkonium chloride. Journal of antimicrobial chemotherapy. 2011 Apr 1;66(4):840-4.

Hesje CK, Borsos SD, Blondeau JM. Benzalkonium chloride enhances antibacterial activity of gatifloxacin and reduces its propensity to select for fluoroquinolone-resistant strains. Journal of ocular pharmacology and therapeutics. 2009 Aug 1;25(4):329-34.

Response 3: We acknowledge that there have been previous publications on increased bacterial killing of ophthalmic fluoroquinolones in the presence of BAK and that such combinations may prevent development of resistant strains. 

Our study expands upon this concept for besifloxacin specifically in the context of concentrations of besifloxacin that might be achievable on eye following topical instillation of an ophthalmic drop.  Note that the previous publication by Haas evaluated a much lower concentration of besifloxacin (namely 0.24 ug/mL), and one that is not as suitable/relevant for modeling the bacterial killing with besifloxacin on eye after topical administration of besifloxacin ophthalmic suspension 0.6%.  The reviewer may wish to review lines 315—324 and 329—340 on this topic for an in-depth discussion.

Point 4: In abstract, line #9 write the concentration units of Bas suspension.

Response 4: We thank the reviewer for this helpful suggestion.  We have added the units to the first line in the Abstract.

Point 5: What is the method used for the activity studies?

Response 5: Current line 14 of the abstract identifies that the method used for the activity studies are time-kill experiments of 180 minutes duration.  No change made.

Point 6: What is the level of significance getting for the data through ANOVA? Write the details in the results section of abstract.

Response 6: We are currently over the allowed word count in the abstract based on journal instructions to authors. 

We defer to the editor whether we should try to work in more details on statistics into the abstract proper.  No change made.

Point 7: In line #38, write the generation of Bas as fluoroquinolone.

Response 7: While there is no formal fluoroquinolone generation designated for besifloxacin, it has a similar chemical structure to, and is often compared with, the 4th-generation fluoroquinolones moxifloxacin and gatifloxacin.  However, it is the only chlorinated ophthalmic fluoroquinolone, which is currently stated in text. No change made..

Point 8: Write the MIC values of Bas, Mox and gati against Gram + pathogens.

Response 8: We have added information on relative MIC values.  See tracked changes on line 44.

Point 9: In case of BAK, write the details on BAK as permeation enhancer along with antimicrobial property.

Response 9: Indeed BAK is known as a permeation enhancer, improving the penetration of other drugs through ocular tissues.  We don’t feel that this information is particularly relevant here, since besifloxacin is indicated for the treatment of bacterial conjunctivitis, an ocular surface disease.  However, please note that the next sentence the introduction speaks to the action of BAK in destabilizing bacterial cell membranes, which is relevant to this paper.  No change made.

Point 10: Line #69, 76, in vitro should be in italics.

Response 10: We defer to the journal editor as to whether this term should be italicized as this is usually a journal style decision.  No change made.

Point 11: Line #103, an inoculum of each isolate – what is the expected colony count of each organism from this inoculum. How to check the consistency of the inoculum usage?

Response 11: Consistency was ensured by Spectrophotometry to confirm the CFU/mL.  We have added this detail to the methods at line 108.  We apologize for our omission.

Point 12: Line #106, 2 hours at 35–37oC, line#107, CO2  – units should be superscript and subscript.

Response 12: Superscripts and subscripts have been fixed.  Thank you for pointing this out.

Point 13: Line #109, cell densities of 104–105 cells/mL – is it too the power of 4 – 5 or 104 – 105 cells/mL.

Response 13: Indeed this should have been 104–105 cells/mL.  The text has  been fixed.  Thank you for pointing this out.

Point 14: What is the reason for selection of testing concentrations of BAK and Bas for the study?

Response 14: Please see Discussion section lines review lines 329—340 for rationale of Besifloxacin concentration and lines 342—353 for discussion on BAK concentrations tested.  No changes made.

Point 15: Line#119, P value of <0.05 – p should be small.

Response 15: We defer to the journal style preference for this.

Point 16: In Fig.1, include SD values and the clarity of the figure is very poor.

Response 16: We have added SD to the figures as requested in cases where there are 3 or more isolates averaged.  Adding  SD in cases where there are fewer isolates is not meaningful.

We agree the clarity of the figures is poor and have reinserted adjusted figures using a different export format into the current manuscript version.  We will also provide original figures files to the journal so that the journal editorial staff can insert the best quality figures.

Point 17: Why 100 ug/mL of BAK in combination with Bas missing in Figure 2.

Response 17: We defer to the journal style preference for this.

Point 18: It is better to split the Table 1 into organism wise. That will useful for the readers for easy comparison.

Response 18: We have added a dividing line after each species to make the table easier to follow.  Thank you for that suggestion.

Round 2

Reviewer 1 Report

Thank you for replying to my comments. In my opinion the manuscript was improved and it is ready for acceptance. 

Beyond review process I'd like to discuss point 5 with authors:

"Point 5: In some panels we see a plateau effect after treatment with BAK (see e.g. S. epidermidis 60-180 minutes with 20 ug/ml BAK or H. influenza (same scenario) however without values of CFU/ml we cannot asses if it was complete eradication of bacteria or it is a classical persisters plateau.

Response 5: As mentioned in the methods, the starting concentration of viable cells was usually 104-105 cells/mL, hence there can be no more bacterial killing once you reach 5 log reductions as there are no more viable organisms to kill. "

Reviewer comment:

This is interesting observation since hardly any antibiotics are able to completely eradicate bacterial population, even in a very high concentrations. As literature have shown, there is always a small persister fraction in a bacterial population. Log reduction are not sufficient to be able to asses if we can see persister cells after the treatment with BAK therefore I'd be really interested to see the CFU/ml number. I also encourage the authors to perform experiments that will answer the question regarding persister population. Good luck!

Reviewer 2 Report

The authors have addressed all the issues raised.

Reviewer 3 Report

No further comments. Thank you for the clarification.